# Antidiabetic Effect of Casein Glycomacropeptide Hydrolysates on High-Fat Diet and STZ-Induced Diabetic Mice via Regulating Insulin Signaling in Skeletal Muscle and Modulating Gut Microbiota

**DOI:** 10.3390/nu12010220

**Published:** 2020-01-15

**Authors:** Qichen Yuan, Biyuan Zhan, Rui Chang, Min Du, Xueying Mao

**Affiliations:** 1Beijing Advanced Innovation Center for Food Nutrition and Human Health, College of Food Science & Nutritional Engineering, China Agricultural University, Beijing 100083, China; cau_yuanqichen@163.com (Q.Y.);; 2Key Laboratory of Functional Dairy, College of Food Science and Nutritional Engineering, Ministry of Education, China Agricultural University, Beijing 100083, China; 3Department of Animal Sciences, Washington State University, Pullman, WA 99164, USA

**Keywords:** casein glycomacropeptide hydrolysate, diabetes, insulin signaling pathway, skeletal muscle, gut microbiota, inflammation

## Abstract

This study evaluated the effects and the underlying mechanisms of casein glycomacropeptide hydrolysate (GHP) on high-fat diet-fed and streptozotocin-induced type 2 diabetes (T2D) in C57BL/6J mice. Results showed that 8-week GHP supplementation significantly decreased fasting blood glucose levels, restored insulin production, improved glucose tolerance and insulin tolerance, and alleviated dyslipidemia in T2D mice. In addition, GHP supplementation reduced the concentration of lipopolysaccharides (LPSs) and pro-inflammatory cytokines in serum, which led to reduced systematic inflammation. Furthermore, GHP supplementation increased muscle glycogen content in diabetic mice, which was probably due to the regulation of glycogen synthase kinase 3 beta and glycogen synthase. GHP regulated the insulin receptor substrate-1/phosphatidylinositol 3-kinase/protein kinase B pathway in skeletal muscle, which promoted glucose transporter 4 (GLUT4) translocation. Moreover, GHP modulated the overall structure and diversity of gut microbiota in T2D mice. GHP increased the *Bacteroidetes*/*Firmicutes* ratio and the abundance of *S24-7*, *Ruminiclostridium*, *Blautia* and *Allobaculum*, which might contribute to its antidiabetic effect. Taken together, our findings demonstrate that the antidiabetic effect of GHP may be associated with the recovery of skeletal muscle insulin sensitivity and the regulation of gut microbiota.

## 1. Introduction

Diabetes is reaching epidemic proportions, with 425 million adults living with it, according to the International Diabetes Federation. Type 2 diabetes (T2D), accounting for more than 90% of diabetes, is a chronic metabolic disease of multifactorial origin, including family history, being overweight, and having an unhealthy lifestyle [1]. T2D is characterized by long-term hyperglycemia and insulin resistance [2], and is usually accompanied with dyslipidemia [3].

Insulin is the main regulator of the blood glucose level by regulating glucose uptake, storage and production in insulin target organs [4]. Skeletal muscle is the primary insulin target tissue, and up to 75% of insulin-mediated glucose uptake occurs in the skeletal muscle at both euglycemia and hyperglycemia [5]. 

Insulin works by binding with insulin receptor (IR), then activated IR recruits insulin receptor substrate-1 (IRS-1) and then phosphorylates it on tyrosine residue. IRS-1 attracts the regulatory subunit p85 of phosphatidylinositol 3-kinase (PI3K) to activate PI3K [6]. Activated PI3K brings about the activation of protein kinase B (Akt), and insulin-stimulated Akt activation inactivates glycogen synthase kinase-3β (GSK-3β). Impaired insulin sensitivity leads to elevated activity of GSK-3β in T2D muscle, thereby inhibiting glycogen synthase (GS) activity and decreasing glucose disposal [7]. Besides, insulin increases glucose uptake through Akt, stimulating the translocation of glucose transporter 4 (GLUT4), the predominant glucose transporter in skeletal muscle, from the cytoplasm to the plasma membrane in muscle cells [8].

The development of T2D is also accompanied with altered gut microbiota composition. Accumulating evidences have indicated that diabetic individuals present different gut microbiota compared with healthy people. The *Firmicutes*/*Bacteroidetes* ratio increases in diabetic people and animals [9,10], and *Proteobacteria* was elevated in diabetic patients compared with normal people [11]. The occurrence of altered gut microbiota results in increased intestinal permeability, and thereby increases LPS absorbed into systemic circulation. Circulating LPS binds to Toll-like receptor 4 (TLR-4), and then activates the inflammatory pathway and leads to insulin resistance [12]. Transplantation of intestinal microbiota from healthy, lean donors improved insulin signaling in participants with metabolic syndrome [13], suggesting that modulating gut microbiota is helpful for ameliorating T2D. Diet supplementations with some natural bioactive ingredients were found to improve glucose metabolism partially by modifying gut microbiome [14,15], providing evidence for alleviating T2D by regulating gut microbiota.

Currently, antidiabetic medicines available for T2D patients have various degrees of side effects, including hypoglycemia, weight gain and gastrointestinal side effect [16], so it is urgent to explore natural bioactive compounds which are safer and more economical. Casein glycomacropeptide (GMP), derived from milk κ-casein, is a glycopeptide composed of 64 amino acid residues. GMP exerts varieties of biological activities. GMP-derived peptide could prevent high glucose-induced insulin resistance in HepG2 cells via activating the IRS/PI3K/Akt signaling pathway [17]. In addition, hydrolysate of GMP (GHP) increased the level of hepatic glycogen and ameliorated the hepatic insulin resistance of high-fat diet (HFD)-fed mice [18]. These studies suggest that GHP may improve the insulin sensitivity of insulin target organs. In our previous study, GHP could reduce the levels of interleukin-6 (IL-6), interleukin-1 beta (IL-1β) and tumor necrosis factor-alpha (TNF-α) in macrophages via Akt mediated nuclear factor-κB (NF-κB) signaling [19], and reduced pro-inflammatory cytokines in HFD-induced obesity rats [20], suggesting that GHP may have benefits on gut microbiota. Based on these results, we hypothesized that GHP could alleviate T2D by recovering insulin sensitivity and modulating gut microbiota. To verify the hypothesis, HFD-fed and streptozotocin (STZ)-induced diabetic C57BL/6J mice were used, and the effects of GHP on skeletal muscle insulin signaling and the gut microbiota were investigated to preliminarily explore the mechanism.

## 2. Materials and Methods

### 2.1. Preparation of GHP

Glycomacropeptide hydrolysate (GHP) was prepared as described previously [19]. Briefly, glycomacropeptide (GMP) (CGMP-20, Arla Foods Ingredients, Viby, Denmark) was dissolved in distilled water at a concentration of 5% (*w*/*v*) and was hydrolyzed using papain with the enzyme-to-substrate mass ratio of 5% at the optimum condition (pH 6.0, 55 °C). The hydrolysis was terminated by heating at 85 °C for 20 min and the hydrolysate was centrifuged. The supernatants were collected and lyophilized to obtain GHP.

### 2.2. Animals and Treatments

Animal study protocol (KY10007) was approved by the Animal Ethics Committee of the China Agricultural University (Beijing, China). Six-week-old male C57BL/6J mice were purchased from the Vital River Laboratory Animal Technology Company Limited. They were housed under a constant temperature (22 ± 1 °C), with a 12-h light/dark cycle. After one-week acclimation, mice were randomly divided into three groups, comprised of the control group, type 2 diabetes (T2D) group and T2D+GHP group (each with *n* = 8). Mice in our control group were fed with common diets (10% calories from fat). To induce diabetes, mice in the T2D and T2D+GHP groups were fed with a high-fat diet (HFD) (60% calorie from fat, D12492, Beijing Keao Xieli Feed Co., Ltd., Beijing, China) for 8 weeks and were given an intraperitoneal injection with streptozotocin (STZ) (Sigma-Aldrich LLC., St. Louis, MO, USA) dissolved in 0.1 M citrate buffer (pH 4.5) at a daily dose of 40 mg/kg bw for five consecutive days in the fifth week. Meanwhile, control mice received an intraperitoneal injection with 3 mL/kg bw 0.1 M citrate buffer (pH 4.5). Fasting blood glucose (FBG) levels were examined in a week after STZ-injection and mice with FBG levels over 11.1 mM were considered to be diabetic. After successful modeling, mice in our T2D+GHP group were intragastrically administrated with additional 200 mg GHP/kg bw, dissolved in physiological saline, for another 8 weeks. Mice in the other two groups were treated with equal volumes of physiological saline and continued on their respective diets. Body weight (BW) and food intake were recorded twice per week.

### 2.3. Serum and Tissue Sample Collection

Mice were anesthetized after overnight fasting. The blood was collected from the orbital venous plexus to collect serum by centrifugation at 1300× *g* at 4 °C for 15 min. Mice were sacrificed by cervical dislocation to resect tissues. The pancreas was fixed in 4% paraformaldehyde solution. The skeletal muscle was quick-frozen by liquid nitrogen and stored at −80 °C.

### 2.4. Fasting Blood Glucose and Serum Insulin

Blood glucose concentration were determined using a Roche glucometer after an overnight fasting. The serum insulin level was determined using an enzyme-linked immunosorbent assay (ELISA) kit (Mercodia, Uppsala, Sweden).

### 2.5. Oral Glucose Tolerance Test (OGTT) and Intraperitoneal Insulin Tolerance Test (IPITT)

The Oral Glucose Tolerance Test (OGTT) and Intraperitoneal Insulin Tolerance Test (IPITT) were conducted after intervention with GHP for 8 weeks. For OGTT, 6-h fasted mice were orally administered glucose at a dose of 2 g/kg bw. For IPITT, 6-h fasted mice were injected intraperitoneally with insulin (#I0305000, Sigma-Aldrich LLC.) at a dose of 1 U/kg bw. Blood was collected from the tail vein at 0, 30, 60, 90 and 120 min after administration, and the blood glucose level was measured with a Roche glucometer. The glucose response was evaluated by area under the curve (AUC).

### 2.6. Plasma Biochemical Analysis

Serum levels of total cholesterol (TC), triglycerides (TG), high-density lipoprotein cholesterol (HDL-C), low-density lipoprotein cholesterol (LDL-C) and free fatty acids (FFA) were determined using commercial kits (Sekisui Medical, Tokyo, Japan). The serum levels of IL-6, IL-1β, TNF-α and the lipopolysaccharides (LPSs) were measured by ELISA kits (R&D Systems, Inc., Minneapolis, MN, USA). All procedures were performed following the manufacturer’s instructions.

### 2.7. Histopathological Analysis

We then fixed the pancreas in 4% paraformaldehyde solution and embedded it in paraffin wax. We then stained the tissues (4 μm thickness) with hematoxylin and eosin (H&E). Histological changes were observed and photographed with an Olympus IX 73 microscope (Olympus Corporation, Tokyo, Japan) at a magnifying power of 200×.

### 2.8. Western Blot Analysis

The Western blotting was performed as previously described [21]. Total proteins of the skeletal muscle were extracted with RIPA lysis buffer (P0013, Beyotime, Haimen, Jiangsu, China). Protein concentration was measured using a bicinchoninic acid (BCA) protein assay kit (PA115, Tiangen Biotech, Beijing, China). The plasma membrane protein of the skeletal muscle was extracted using a Membrane Portein Extraction Reagent (89842, Thermo Fisher Scientific, Waltham, MA, USA). An equal amount of proteins was separated by 10% SDS-PAGE and then transferred to polyvinylidene fluoride (PVDF) membranes (Millipore, Billerica, MA, USA). Blocking the membranes with Tris-buffer saline with 0.1% Tween 20 (TBST) containing 5% skim milk for 1 h, and then incubating membranes with primary antibodies against p-IRS-1 (#2381), IRS-1 (#3407), p-PI3K p85 (#), PI3K p85 (#4257), p-Akt (#4060), Akt (#4691), p-GSK-3β (#9323), GSK-3β (#12356), p-GS (#3891), GS (#3886), GLUT4 (#2213), Na, K-ATPase (#23565), β-tubulin (#2146) (1:1000, Cell Signaling Technology, Danvers, MA, USA) at 4 °C overnight. After washing five times with TBST, incubating membranes with secondary antibody (#7074, Cell Signaling Technology) for 1 h. Enhanced chemiluminescence (ECL) reagents (WBKLS0500, Millipore) were used to visualize protein bands. Images were obtained using Amersham Imager 600 and the protein density was quantified via ImageJ software.

### 2.9. Sequencing and Analysis of Bacterial 16S rRNA Genes

Fresh fecal samples were collected and stored at −80 °C prior to analysis. Fecal genomic DNA was extracted from feces using an E.Z.N.A ^®^ Soil DNA Kit (Omega Bio-tek, Inc., Norcross, GA, USA) according to the manufacturer’s instructions. The V3-4 region of 16S rRNA genes was amplified from extracted DNA by PCR using primers 338F: 5′-ACTCCTACGGGAGGCAGCAG-3′ and 806R: 5′-GGACTACHVGGGTWTCTAAT-3′. The PCR reaction (20 μL) contained 10 ng of DNA, 4 μL of 5× Fast Pfu Buffer, 2 μL of 2.5 mM dNTPs, 0.8 μL of each primer (5 μM), and 0.4 μL of Fast Pfu Polymerase. After amplification, sequencing was performed by Majorbio Bio-Pharm Technology Co. Ltd. (Shanghai, China), using the Illumina Sequencer Miseq platform as previously described [22]. Similar sequences were binned into operational taxonomic units (OTUs) using UPARSE (version 7.1), with a 97% similarity threshold. The Simpson and Shannon indices were calculated in QIIME (version 1.17) to assess community diversity. Principal coordinate analysis (PCA) was performed according to the distance matrices created by QIIME. To identify significantly different species at the OTU level, linear discriminant analysis (LDA) effect size (LEfSe) analysis was conducted, and the discovered species with an LDA score higher than three were more abundance in the respective group than the other two groups. 

### 2.10. Statistical Analysis

Results are represented as the mean ± standard error of the mean (SEM). The statistical analysis was conducted using the SPSS software (version 19.0, Inc, Chicago, IL, USA). Significant differences between groups were assessed by one-way analysis of variance (ANOVA) with Duncan multiple comparisons. Differences were considered to be statistically significant with *p* value < 0.05.

## 3. Results

### 3.1. Effects of GHP on Body Weight, Fasting Blood Glucose and Serum Insulin Levels in T2D Mice

BW increased faster in T2D and T2D+GHP groups than that of the control group in the first four weeks due to HFD feeding. After the injection of STZ in the fifth week, the BW decreased significantly in the T2D group and in the T2D+GHP group (Appendix A). However, GHP administration significantly recovered the loss of BW caused by STZ without affecting food intake (Figure 1A,B). 

At the end of the experiment, STZ-induced hyperglycemia was notably reduced after the administration of GHP in the T2D+GHP group (*p* < 0.01) (Figure 1C). In addition, the insulin level in T2D was effectively reduced compared with the control group. GHP intervention could significantly increase the serum insulin level compared with the T2D group (Figure 1D).

### 3.2. Effects of GHP on OGTT and IPITT in T2D Mice

The blood glucose levels reached the maximum peak at 30 min after the glucose gavage in all groups during the OGTT, and the peak value of glucose was higher in the T2D group and T2D+GHP group compared to that of the control group. The blood glucose levels gradually recovered to near normal levels over the next 90 min in the control group while they remained at a high level in the T2D group. Administration of GHP showed significant decrease in blood glucose concentration over the next 90 min in the T2D+GHP group as compared to the T2D group (Figure 2A). Glucose AUC was higher in T2D group and treatment with GHP obviously reduced the AUC compared with the T2D group (Figure 2B). The blood glucose levels dropped to the minimum at 60 min in all groups, and they were higher in the T2D group than in the control group during the IPITT. After the administration of GHP, the glucose level was notably reduced compared with that in the T2D group (Figure 2C). Similarly, the AUC of glucose in IPITT was obviously higher in the T2D group and was notably lowered in the T2D+GHP group (Figure 2D).

### 3.3. Effects of GHP on Plasma Lipid Levels in T2D Mice

The T2D group showed a conspicuous higher concentration of serum TG, LDL-C and FFA, while these lipid levels were significantly lowered by the administration of GHP in the T2D+GHP group. Moreover, GHP administration significantly increased serum HDL-C levels in the T2D+GHP group as compared to the T2D group. However, no significant difference was found in the concentration of serum TC between the T2D group and the T2D+GHP group (*p* > 0.05) (Table 1).

### 3.4. Effects of GHP on Serum Inflammatory Biomarkers in T2D Mice

The TNF-α, IL-1β, IL-6 and LPS levels were notably elevated in T2D mice compared to control mice. The administration of GHP dramatically decreased these serum inflammatory biomarkers in our T2D+GHP group (Table 1).

### 3.5. Histopathological Analysis

The pancreatic tissues of mice in the control group displayed a clear and normal structure of the islets of Langerhans, which contained abundant and full form islets. In contrast, the pancreatic tissues in the T2D group were injured and showed a distortion general architecture of the islets of Langerhans, which contained few irregular and swelling islets. However, the islets of Langerhans were significantly improved in the T2D+GHP group as compared to those in the T2D group (Figure 3).

### 3.6. GHP Promoted Glycogen Synthesis in T2D Mice

The content of muscle glycogen was evidently lower in T2D mice than the control mice, but supplementation of GHP significantly elevated glycogen storage in the T2D+GHP group (Figure 4A). As expected, the level of p-GSK-3β was markedly lowered in T2D mice than that of the control mice, and 8-week administration of GHP increased the level of p-GSK-3β as compared with the T2D group, indicating that GSK-3β was deactivated. Accordingly, the level of p-GS was evidently increased in T2D mice compared with control mice, and GHP supplementation decreased it significantly (Figure 4B,C).

### 3.7. GHP Elevated Insulin Signaling Pathway and Promoted GLUT4 Translocation in the Skeletal Muscle in T2D Mice

Key protein expression levels of the insulin signaling pathway in the skeletal muscle were assayed. Results showed that the p-IRS-1 (Ser307) of the T2D group increased significantly (*p* < 0.01), and the p-PI3K/PI3K and p-Akt/Akt ratios were markedly decreased in the T2D group compared with the control group. After the administration of GHP, these changes were significantly attenuated (Figure 5A,B).

The predominant glucose transporter in skeletal muscle is GLUT4, and the translocation of GLUT4 depends in part on insulin-mediated signaling pathways. As compared with the control mice, the expression level of GLUT4 in the plasma membranes of skeletal muscle decreased by 32.5% in the T2D group. However, GHP promoted GLUT4 translocation from cytoplasm to the plasma membrane in diabetic mice (Figure 5C,D).

### 3.8. GHP Modulated the Composition and Diversity of the Gut Microbiota in T2D Mice

At the phylum level, the gut microbiota of T2D mice displayed significant decrease in the abundance of Bacteroidetes and increases in Firmicutes and Proteobacteria compared with the control group. GHP supplementation significantly reversed the abundance change of Bacteroidetes and Proteobacteria, and increased the *Firmicutes*/*Bacteroidetes* ratio compared with the T2D group (Figure 6A,B). At the family level, gut bacteria of the T2D group displayed increased abundance of *Helicobacteraceae*, *Lachnospiraceae* and *Lachnospiraceae*, and decreased abundance of *Bacteridaless_S24-7_group* compared with the control group. However, a reduced abundance of *Helicobacteraceae* and an increased abundance of *Ruminococcaceae* and *Bacteroidales_S24-7_group* were found in the T2D+GHP group compared with the T2D group (Figure 6C,D). PCA showed that the gut communities among groups separated markedly with the PC1 percent variation explained as being equal to 42.69%, and the PC2 percent variation was explained as being equal to 20.59% (Figure 6E). Besides, the Simpson index was higher and the Shannon index was lower in the T2D group than those of the control group, suggesting that the community diversity was decreased by HFD and STZ treatment. However, GHP significantly recovered the two indices in comparison with the T2D group (Figure 6F,G).

### 3.9. GHP on Taxonomic Diversity of the Gut Microbiota in T2D Mice

To further investigate the taxonomic diversity modulated by GHP, LEfSe was used to compare the bacteria composition of the three groups. As shown in Figure 7, there was a higher abundance of *Bacteroidetes* in the normal mice than in the T2D mice. HFD and STZ treatment significantly increased in 15 genera of *Firmicutes*, *Proteobacteria* and *Deferribacteres*. GHP supplementation induced 13 higher genera belonged to *Actinobacteria*, *Bacteroidetes*, *Firmicutes* and *Proteobacteria*, including *Anaerovorax*, *Blautia*, *Coprococcus_3*, *Anaerotruncus*, *Ruminiclostridium*, *Ruminiclostridium_9*, *Ruminiclostridium_5*, *norank_f_Peptococcaceae*, *Allobaculum*, *Butyricimonas*, *Rikenella*, *Burkholderia_Paraburkholderia* and *Coriobacteriaceae_UCG_002.*

### 3.10. Correlation between Gut Microbiota and T2D-Related Parameters

Spearman’s correlation analysis was performed to predict the correlations between gut microbiota (40 major genera) and T2D-related metabolic parameters. As shown in Figure 8 the heatmap reflected significant positive correlations between antidiabetic effects and *Parasutterella* and *Prevotellaceae_UCG-001*. *Anaerotruncus*, *Bilophila*, *Fecalibaculum*, *Helicoacter*, *Intestinimonas*, *Mucispirillum*, *Romboutsia*, and the *Eubacterium_xylanophilum_group* were negatively correlated with hypoglycemic, hypolipidemic, anti-inflammation and glycogen synthesis effects (≥9 asterisks).

## 4. Discussion

The pathological characteristics of T2D are complicated, including pancreas β-cells damage, decreased insulin sensitivity in target organs, inflammation, gut microbiota dysbiosis and immune disorders [2], and these are also potential therapeutic targets for T2D. The pancreatic β-cell is responsible for insulin secretion and blood glucose homeostasis. Impaired pancreatic β-cell function contributes to the development of T2D [23]. Therefore, restoring β-cell function is considered to be a feasible therapy for T2D [24,25]. In the present study, an HFD-fed and STZ-induced diabetic mouse model with β-cell dysfunction was established considering STZ selectively damages β-cell via entering β-cell through glucose transport 2 and causing β-cell DNA methylation [26]. Eight weeks of GHP administration restored β-cell function, as evidenced by decreased FGB, increased serum insulin levels and reversed pancreatic islet lesion (Figure 1 and Figure 3). Based on the beneficial effects of GHP on insulin secretion, effects of GHP on peripheral insulin sensitivity was investigated.

Skeletal muscle is the primary tissue of insulin-mediated glucose disposal, playing a crucial role in glucose metabolism. After transporting into muscle cells, glucose converts to glycogen. Glycogen synthesis is dependent on the activity of GS and glucose uptake. GS activity is regulated by GSK-3β, which is modulated by the upstream insulin signaling pathway. In skeletal muscle, glucose uptake is mainly mediated by GLUT4, which is stored in intracellular vesicles and can rapidly translocate to the cell membrane stimulated by insulin via the PI3K/Akt pathway [27]. Therefore, activating muscle insulin signaling could be an effective treatment of T2D. In the state of insulin resistance, IRS-1 serine phosphorylation inhibits insulin signaling by decreasing tyrosine phosphorylation, and thereby inhibits downstream effectors in insulin signal transduction, such as PI3K, Akt and GSK-3β [4]. Therefore, reducing serine phosphorylation to increase the tyrosine phosphorylation of IRS-1 is a target for improving insulin sensitivity. Unacylated ghrelin restored impaired IRS/Akt signaling and enhanced GLUT4 translocation in diabetic muscle, contributing to its antidiabetic effect [28]. Tocotrienol-rich fraction supplementation upregulated IRS-1 and Akt levels with increased translocation of GLUT4 in skeletal muscle, showing the antidiabetic effect in T2D mice [29]. In the present study, supplementation of GHP induced a decreased expression of p-IRS-1 and increased expressions of p-PI3K and p-Akt, and elevated the muscle glycogen level via increasing glucose uptake mediated by GLUT4 and increasing GS activity in the skeletal muscle in T2D mice, indicating that the antidiabetic effect of GHP was partly via restoring the insulin signaling pathway.

It has been proven by fecal transplantation experiments that gut microbiota can directly affect insulin sensitivity [13], suggesting that the modulation of gut microbiota is an effective way to ameliorate T2D. *Firmicutes* and *Bacteroidetes* account for 80–90% of the bacterial phylotypes [30]. The imbalance between *Firmicutes* and *Bacteroidetes* is associated with the development of diabetes [11,31]. *Proteobacteria*, which is another predominant phyla in gut microbiota, contains many pathogens [32], and was elevated in diabetic patients compared with normal people [11]. Thus, the decrease in the abundance of *Proteobacteria* might be associated with the alleviation of T2D. Administration of *Alpinia oxyphylla* extract showed antidiabetic effect partially via decreasing the *Firmicutes/Bacteroidetes* ratio in db/db mice [33]. *Potentilla discolor* bunge wat extract exerted an antidiabetic effect via reducing the *Firmicutes/Bacteroidetes* ratio and the abundance of *Proteobacteria* in diabetic mice [34]. In this study, GHP prevented the loss of *Bacteroidetes* and suppressed the increase of *Firmicutes* and *Proteobacteria* in T2D mice, suggesting that GHP may exert its antidiabetic effects via regulating the integral structure of gut microbiota. 

The underlying mechanism by which microbiota affects insulin sensitivity may be metabolic inflammation [12]. The *Bacteroidales* family *S24-7* showed negative correlations to blood glucose, dyslipidemia and inflammation [22,35], and positive correlation to PI3K [36]. Besides, most phyloytpes in *Allobaculum* and *Blautia* can produce short-chain fatty acids (SCFAs), which were reported to enter the systemic circulation and be absorbed by muscle cells, and work in alleviating inflammation and promoting insulin sensitivity [37,38,39]. Besides, SCFA promotes the release of glucagon-like eptide-1, which increases GLUT4 gene expression and glucose uptake in skeletal muscle [40]. Tea water extracts attenuated HFD-induced metabolism disorders and modulated gut microbiota in mice with increased abundance of *Ruminococcaceae* [41]. 

However, undesirable changes in gut microbiota composition aggravate inflammation. For example, *Helicobacteraceae* triggers inflammatory and immune response and influences glucose and lipids absorption. *Helicobacter pylori* treatment could improve the glycosylated hemoglobin level in T2D patients [42]. The undesirable changes in gut microbiota elevates intestinal permeability and increases the circulating levels of LPS, which further triggers insulin resistance and the production of pro-inflammatory factors [12]. LPS inhibits insulin gene expression in a TLR-4 dependent way and via NF-κB signaling in human pancreatic islets [43]. Low muscle glycogen levels were accompanied by increased IL-6 which could induce insulin resistance [40]. Therefore, modulating gut microbiota is an effective way to restore insulin sensitivity. Mulberry fruit polysaccharide and metformin treatment was found to notably increase *Allobaculum* in diabetic mice [44]. *Alpinia oxyphylla* suppressed the growth of *Helicobacter* in T2D mice, contributing to its beneficial effect on ameliorating T2D [33]. According to the studies mentioned above, GHP may promote insulin sensitivity in skeletal muscle and exert antidiabetic effect due to the up-regulation of beneficial bacteria, including *S24-7*, *Ruminiclostridium*, *Blautia* and *Allobaculum* and the down-regulation of the abundance of *Helicobacteraceae*.

Based on these findings, our results provide compelling evidence that the underlying mechanism of the antidiabetic effect of GHP may be restoring insulin sensitivity via activating insulin signaling pathway in skeletal muscle and regulating gut microbiota composition, and the Spearman’s correlation analysis further confirmed this conclusion. Although our results are encouraging, the specific mechanism of GHP on T2D requires further exploration through cell experiments.

## 5. Conclusions

In summary, GHP displayed effective hypoglycemic activity, as well as ameliorated dyslipidemia and inflammation in HFD and STZ-induced diabetic mice. Insulin-mediated glucose uptake and glycogenesis in the skeletal muscle was recovered via the activating of the IRS-1/PI3K/Akt pathway. Furthermore, supplementation of GHP could change the composition and diversity of gut microbiota, which may contribute to the beneficial effects of GHP on the insulin signaling pathway and host metabolism. This study indicates that GHP possesses potential effects on the prevention and management of T2D.

## Figures and Tables

**Figure 1 nutrients-12-00220-f001:**
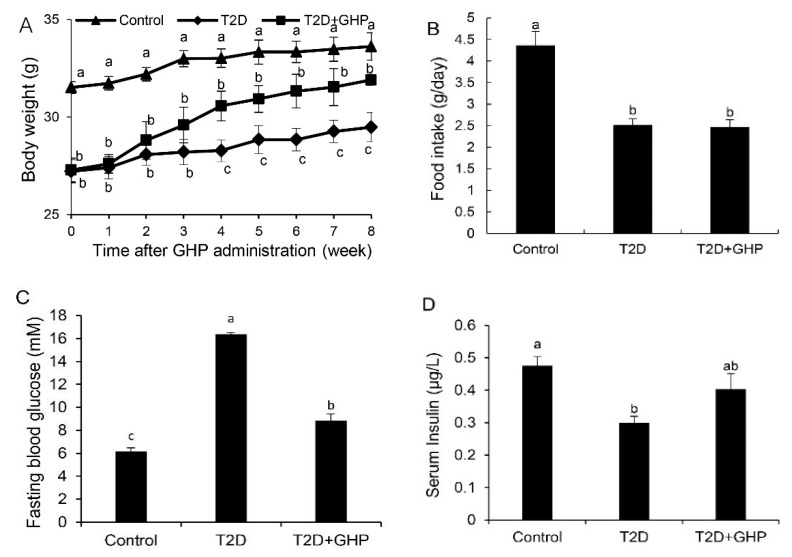
Glycomacropeptide hydrolysate (GHP) supplementation ameliorated body weight, fasting blood glucose (FBG) and serum insulin levels of type 2 diabetes (T2D) mice. (**A**) Body weight during GHP administration. (**B**) Food intake during GHP administration. Control group was fed with normal diet; T2D and T2D+GHP groups were fed with high-fat diet. FBG levels (**C**) and fasting serum insulin levels (**D**) after two-month GHP administration. Results were expressed as means ± the standard error of the mean (SEM) (*n* = 8). Different letters (a–c) indicate significant difference (*p* < 0.05).

**Figure 2 nutrients-12-00220-f002:**
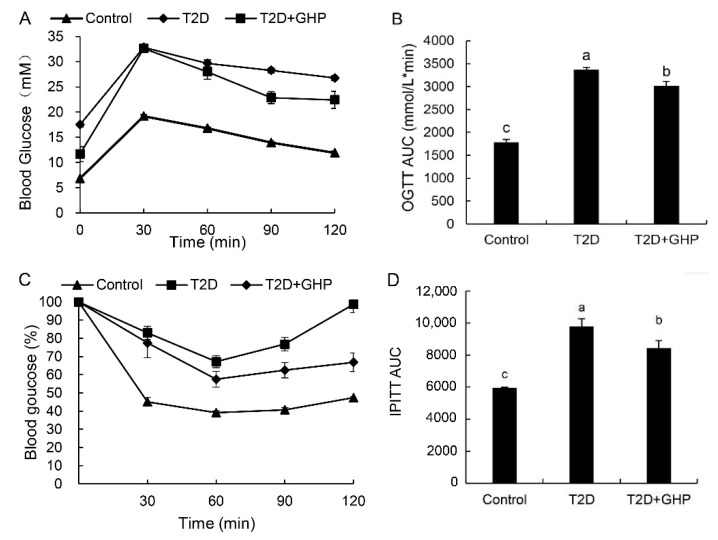
GHP supplementation recovers glucose tolerance and insulin tolerance in T2D mice. Changes in glucose levels during Oral Glucose Tolerance Test (OGTT) (**A**) and the Intraperitoneal Insulin Tolerance Test (IPITT) (**C**). The area under the curve (AUC) of the glucose level during the OGTT (**B**) and IPITT (**D**). Results were expressed as means ± SEM (*n* = 8). Different letters (a–c) indicate significant difference (*p* < 0.05).

**Figure 3 nutrients-12-00220-f003:**
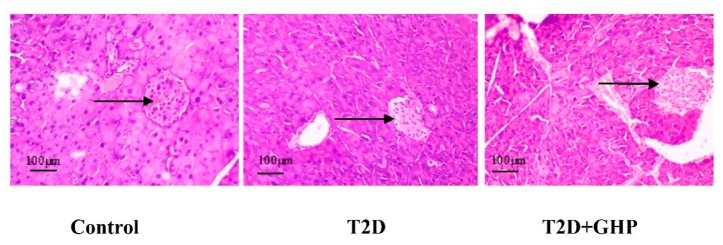
Hematoxylin and eosin (H&E) staining images of the pancreas. Black arrows indicated the islet. Magnification is 200×.

**Figure 4 nutrients-12-00220-f004:**
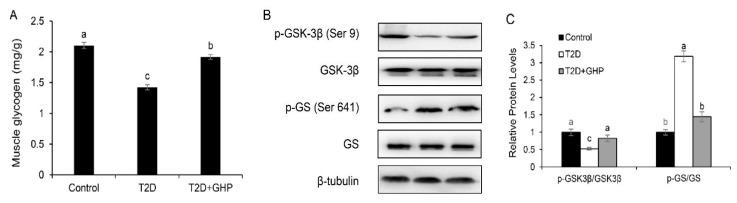
GHP supplementation promotes glycogen synthesis in T2D mice. (**A**) Skeletal muscle glycogen content. (**B**) Protein contents of GSK-3β, p-GSK-3β, p-GS and GS in skeletal muscle and (**C**) quantification data. Protein contents were adjusted to β-tubulin as the loading control. Results were expressed as means ± SEM, *n* = 8. Different letters (a–c) indicate significant difference (*p* < 0.05).

**Figure 5 nutrients-12-00220-f005:**
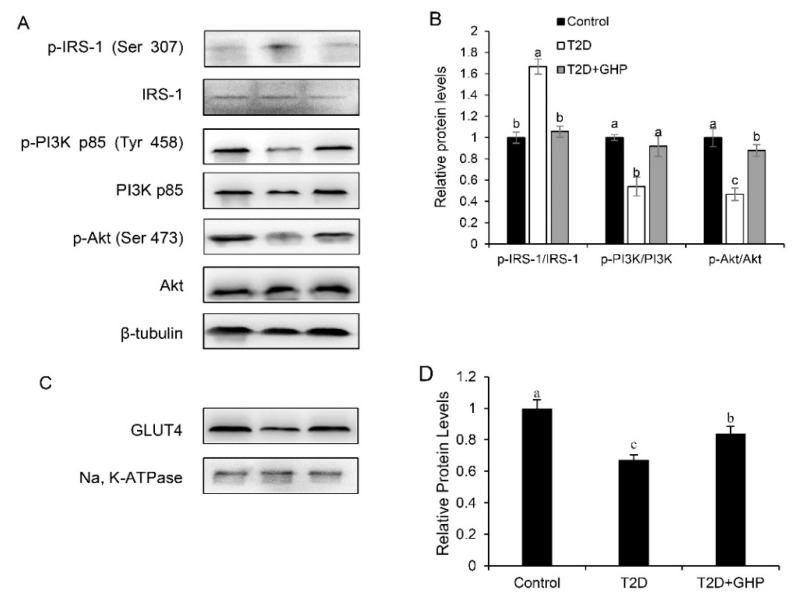
GHP supplementation activates the IRS-1/PI3K/Akt pathway and stimulates glucose transport in T2D mice. (**A**) Skeletal muscle protein contents of IRS-1, p-IRS-1, PI3K, p-PI3K, Akt and p-Akt and (**B**) quantification data. (**C**) Skeletal muscle plasma membrane protein contents of GLUT4 and (**D**) quantification data. Protein contents of IRS-1, p-IRS-1, PI3K, p-PI3K, Akt and p-Akt were adjusted to β-tubulin as the loading control, and content of GLUT4 was adjusted to Na, K-ATPase as the loading control. Results were expressed as means ± SEM, *n* = 8. Different letters (a–c) indicate significantly difference (*p* < 0.05).

**Figure 6 nutrients-12-00220-f006:**
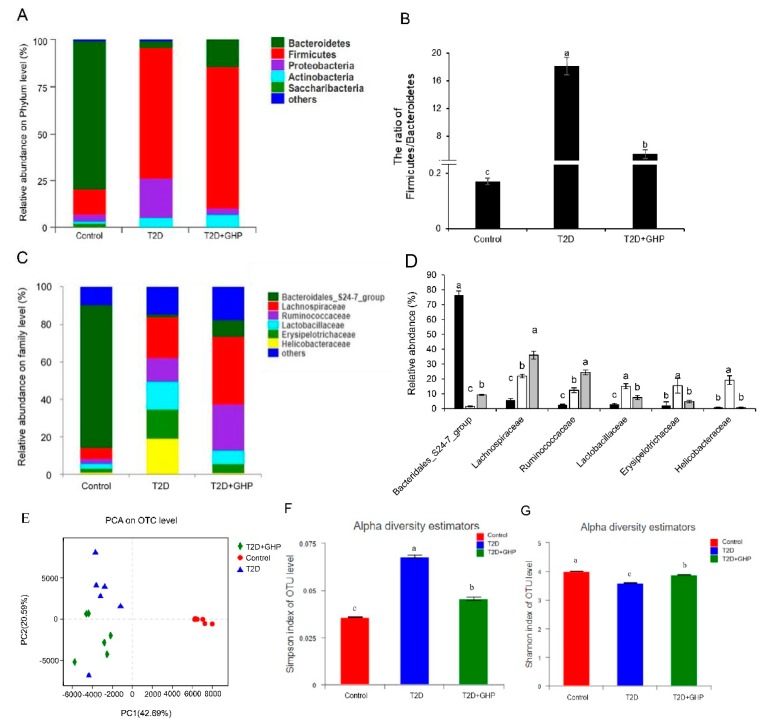
GHP supplementation regulated the composition and diversity of the gut microbiota in T2D mice. (**A**) Relative abundance of gut microbiota at the phylum level. (**B**) The ratio of *Firmiutes*/*Bacteroidetes*. (**C**,**D**) Relative abundance of gut microbiota at family level. (**E**) PCA plots at OTU level. (**F**) Simpson index and (**G**) Shannon index at OTU level.

**Figure 7 nutrients-12-00220-f007:**
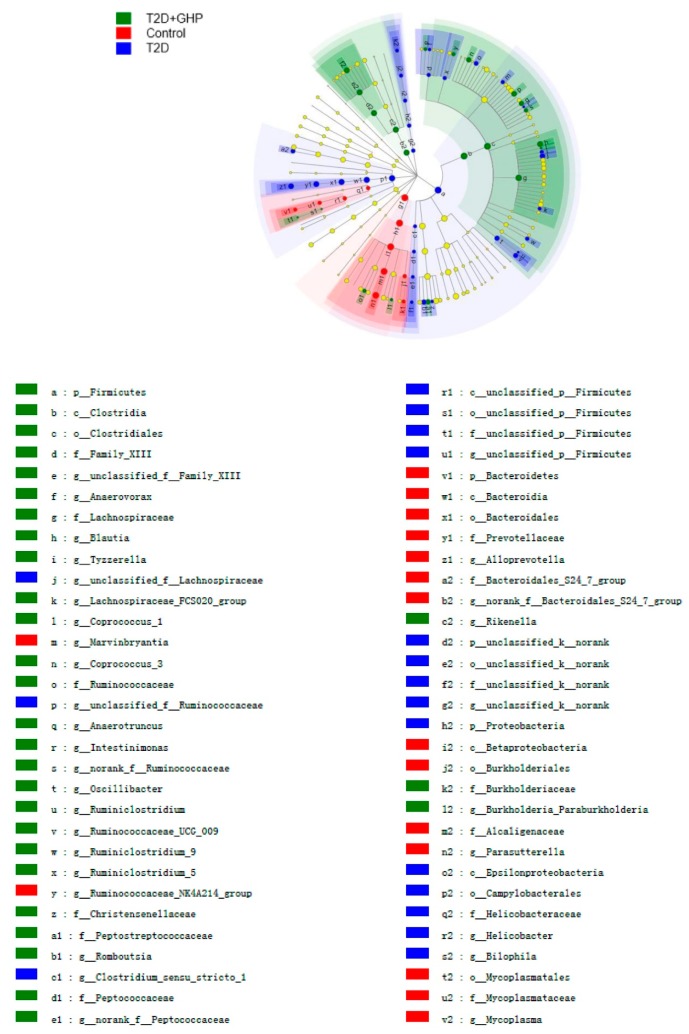
GHP supplementation altered the taxonomic diversity of gut microbiota in T2D mice. Indicator bacteria with LDA scores of 3 or greater are listed. Different color nodes represent microbial groups that are significantly enriched in corresponding groups and have significant changes. Non-significant changes are marked yellow. Each circle is proportional to the abundance of the group. Letters in the figure are noted in the legend.

**Figure 8 nutrients-12-00220-f008:**
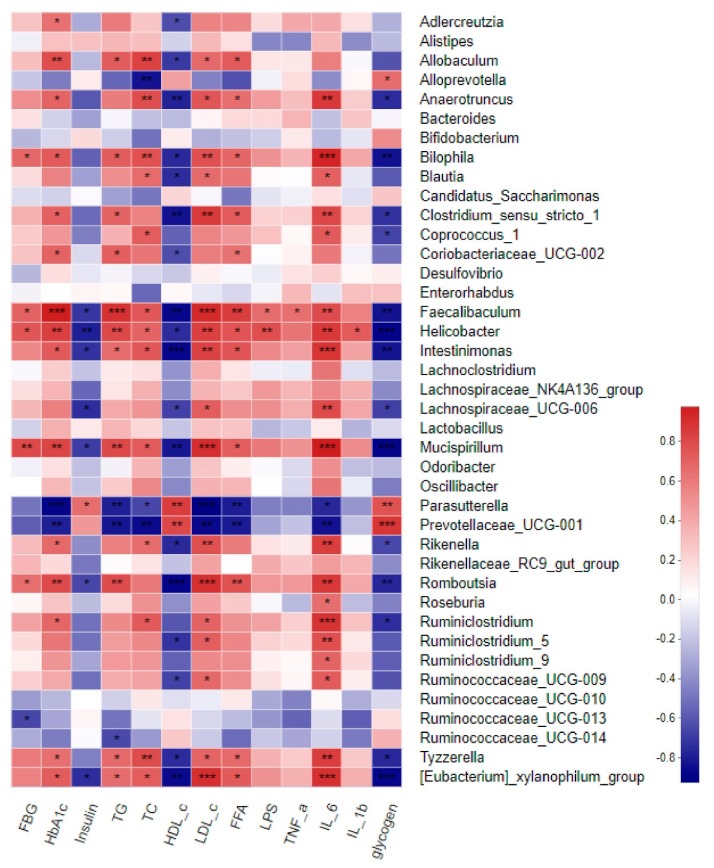
Heatmap of Spearman’s correlation between gut microbiota and T2D-related metabolic parameters. Red color represents a positive correlation, while blue color represents a negative correlation. * 0.01 < *p* ≤ 0.05, ** 0.001 < *p* ≤ 0.01, *** *p* ≤ 0.001.

**Table 1 nutrients-12-00220-t001:** Effects of GHP supplementation on the serum lipid and inflammation in T2D mice.

Parameters	Control	T2D	T2D+GHP
TG (mM)	0.86 ± 0.04c	2.95 ± 0.04a	2.06 ± 0.02b
TC (mM)	2.81 ± 0.09b	4.67 ± 0.05a	4.38 ± 0.10a
HDL-C (mM)	1.64 ± 0.06a	1.14 ± 0.03c	1.31 ± 0.02b
LDL-C (mM)	1.04 ± 0.04c	1.88 ± 0.04a	1.67 ± 0.03b
FFA (mM)	0.92 ± 0.05c	3.76 ± 0.05a	2.93 ± 0.05b
IL-6 (pg/mL)	24.54 ± 2.80c	109.60 ± 2.22a	56.96 ± 2.39b
IL-1β (pg/mL)	15.27 ± 1.01b	25.26 ± 1.57a	16.44 ± 1.05b
TNF-α (pg/mL)	36.77 ± 2.834c	62.50 ± 4.61a	53.40 ± 2.81b
LPS (pg/mL)	277.50 ± 11.30b	609.53 ± 26.74a	285.313 ± 8.11b

Results were expressed as means ± SEM (*n* = 8). Different letters (a–c) indicate significant difference (*p* < 0.05).

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
