# Peer review of "Antidiabetic Effect of Casein Glycomacropeptide Hydrolysates on High-Fat Diet and STZ-Induced Diabetic Mice via Regulating Insulin Signaling in Skeletal Muscle and Modulating Gut Microbiota"

_nutrients, 2020, doi:10.3390/nu12010220_

Round 1
Reviewer 1 Report
The authors have followed my recommendations, however, I continue having some concerns.
Something occurred with figure 1. And please, indicate in Figure 1B that food intake is of different types of diets.
About the LEFSE analysis the authors are not using only species level bacteria. How do you used the LEFSE analysis? Because it is not implemented within QIIME
Table 1. Please, in TC levels change the significant c by a b.
Data in 3.8 is only qualitative or also statistically relevant? Because it is needed to mention it and provide the values. Especially important for figures 6C and 6D.
Please, don’t generalize with “Increased abundance of Firmicutes enhances the capacity to harvest energy from the diet, which induces insulin resistance [28], while higher abundance of Bacteroidetes acts in favor of T2D protection [29].” Some of the most beneficial bacteria belongs to Firmicutes phylum, in fact, the authors use this argument afterword.
It is very interesting the relationship found between Faecalibacterium and LPS. Why Faecalibacterium increases LPS? It has been categorized as a beneficial bacteria and it is a Gram positive bacteria.
Why haven’t the authors been correlated bacteria with skeletal muscle values? There are some reports that have established the relationship between muscle and gut microbiota.
The authors have followed my recommendations, however, I continue having some concerns.
Something occurred with figure 1. And please, indicate in Figure 1B that food intake is of different types of diets.
About the LEFSE analysis the authors are not using only species level bacteria. How do you used the LEFSE analysis? Because it is not implemented within QIIME
Table 1. Please, in TC levels change the significant c by a b.
Data in 3.8 is only qualitative or also statistically relevant? Because it is needed to mention it and provide the values. Especially important for figures 6C and 6D.
Please, don’t generalize with “Increased abundance of Firmicutes enhances the capacity to harvest energy from the diet, which induces insulin resistance [28], while higher abundance of Bacteroidetes acts in favor of T2D protection [29].” Some of the most beneficial bacteria belongs to Firmicutes phylum, in fact, the authors use this argument afterword.
It is very interesting the relationship found between Faecalibacterium and LPS. Why Faecalibacterium increases LPS? It has been categorized as a beneficial bacteria and it is a Gram positive bacteria.
Why haven’t the authors been correlated bacteria with skeletal muscle values? There are some reports that have established the relationship between muscle and gut microbiota.
Reviewer 2 Report
The authors made revisions, and these changes are well-received. But the reviewer still has some concerns on the contents.
I understand T2D and T2D+GHP groups were fed with HFD for 8 weeks, STZ injection was performed in the fifth week, and fed with common diets from the ninth week. As another reviewer has already pointed out, I am also afraid control and T2D/T2D+GHP groups couldn't be compared. I suggest saline-injected mice were used as control.
As the authors presented, serum insulin levels were increased after GHP administration in T2D mice. Authors should also be discussed on the effects of GHP regarding the restoration of beta cell function.
In Figure 5, authors showed GHP promoted GLUT4 translocation in the skeletal muscle, however, it remains unclear whether GHP increased glucose uptake, which should be revised.
Reviewer 3 Report
The authors have made the necessary corrections, but a few issues are still unclear.
Please give the information that during induction of diabetes with STZ, the control group was injected with 0.01M citrate buffer. Why did you remove the information about the percentage of calories that were provided by the high-fat diet (60% calorie from fat)?
I also concern about details of diabetes induction, are the authors sure that high fat diet feeding in the initial stage lasted 8 weeks. According to figures of body mass gain during experiment it lasted 6 weeks (pages 5 and 6).
Please delete the figure on the bottom of page 5.
Were correlation coefficients calculated for all rats?
I have the impression that not all abbreviations have been explained in the manuscript text, e.g. GS - glycogen synthase. A good solution would be to create a list of abbreviations used.
Round 2
Reviewer 1 Report
The authors have followed my comments and recommendations, so I consider that the manuscript could be accepted.
Reviewer 3 Report
No additional comments.
This manuscript is a resubmission of an earlier submission. The following is a list of the peer review reports and author responses from that submission.
Round 1
Reviewer 1 Report
The manuscript by Qichen Yuan et al entitled “Anti-diabetic effect of casein glycomacropeptide hydrolysates on high-fat diet and STZ-induced diabetic mice via regulating insulin signaling in skeletal muscle and modulating gut microbiota” is an interesting report about the use of natural products to reduce diabetic issues. However, some problems with the design of the study and the lack of a proper interconnection model does not permit to achieve a significant conclusion. The lack of a proper control group with HFD does not permit to make correct conclusions about gut microbiota. The control group should be removed. However, sample size is far too small to reach significant conclusions.
Some concerns:
Introduction
Introduction is too long and has too much general information. It should be reduced.
Which is the connection between probiotics and metformin? Are the authors comparing both interventions? Please, reformulate the introduction of antidiabetic drugs.
“However, whether GHP has anti-diabetic effect in HFD and STZ-induced diabetic mice remains unclear”. I don’t think this is the more interesting approach to explain the aim of the study.
Methods
If I’ve understood well, “normal” mice were divided in three groups (sample size?): control, T2D and T2D+GHP. T2D and T2D+GHP groups were fed with HFD and were injected STZ. 9thweek is considered as the starting point. If in this manner, control and diabetic groups couldn’t be compared, at least, about microbiota analysis, as diet is the factor which exerts the greatest influence on gut microbiota.
About microbiota analysis, please, there are multiple typos about OTUs (changed for OUTs). Please, amend.
Results
In general, please, include the sample size. I can see in the PCA plot that sample size is around 5-6 mice. Too small to achieve statistical assumptions.
Figure 1. figures should be reformulated with the 9thweek as the starting point of the study. Previous weeks are not part of the experiment. Authors can include them as supporting material.
About correlations, the whole population was used?
Discussion
If authors feel that gut microbiota is important in the model that they have proposed, it should be more interconnections as this is the main part of the article and the novel one. However, this reviewer continues taking some concerns about the design of the groups to be able to respond the study hypothesis.
Reviewer 2 Report
Yuan et al. addressed the anti-diabetic effects of GHP in high fat diet-fed and STZ-induced diabetic mice. In the diabetic mice, the GHP administration could decrease fasting blood glucose levels concomitant with an amelioration of glucose intolerance and dyslipidemia. These results may imply the beneficial effects of GHP on the prevention and management of type 2 diabetes. The concept of this manuscript is of interest; however, this work is just descriptive not mechanistic.
Major comments
No direct association was shown between the GHP administration and an improvement of glucose metabolism in vivo. Whether the recovery of skeletal muscle insulin sensitivity and the regulation of gut microbiota is the cause or consequence of improved glucose tolerance? Which molecules were involved in the enhancement of insulin signaling pathway in the skeletal muscle and in the modulation of gut microbiota? It may also be necessary to evaluate glucose uptake in the skeletal muscle. In relation to 1, it may be necessary to evaluate the number of islets in the pancreas and/or basal and glucose-stimulated insulin secretion in the islet of T2D+GHP mice because the serum insulin levels were reduced after GHP administration in the diabetic mice as shown in Figure 1D.
Minor comments
In Figures 2C-D, please show the result of ITT as a relative percentage value compared to the value of blood glucose in time 0. Results of Figures 1B and 1D should be described. There is a contradiction on the timing of STZ injection between in the Methods and in the Results, which should be corrected.Reviewer 3 Report
The paper discusses the hypoglycemic effect of casein glycomacropeptide hydrolysates and explains the effect of this compound on insulin signaling. The modulating effect of GHP on microbiota has also been described.
Appropriate research methods were used, but they should be detailed. The ethics committee approval number is missing. The methodological part also does not specify how was GHP administered to mice. Although under the description of Figure 1 states that GHP was administered intragastrically, in my opinion, this information should be given in the methodological section as well.
In the “Results”, the first sentence describing the effect of T2D on body weight should be modified, indicating that the increase in body weight in the T2D and T2D + GHP groups in the first four weeks of experiment is caused by a high fat diet feeding.
In this part in the text there is no reference to the figure 1 and the description of the effect of GHP on fasting insulin levels is missing.
In the table 1 there is incorrect information about differences between groups (T2D vs T2D + GHP) for total cholesterol (TC), because in the description of the results it was depicted that there were no differences between those groups.